# The New Treatment Methods for Non-Hodgkin Lymphoma in Pediatric Patients

**DOI:** 10.3390/cancers14061569

**Published:** 2022-03-18

**Authors:** Justyna Derebas, Kinga Panuciak, Mikołaj Margas, Joanna Zawitkowska, Monika Lejman

**Affiliations:** 1Student Scientific Society, Laboratory of Genetic Diagnostics, Medical University of Lublin, A. Racławickie 1, 20-059 Lublin, Poland; justyna.derebas@gmail.com (J.D.); kinga.panuciak26@gmail.com (K.P.); mikolajmargas@interia.pl (M.M.); 2Department Hematology, Oncology and Transplantology, Medical University of Lublin, A. Racławickie 1, 20-059 Lublin, Poland; jzawitkowska1971@gmail.com; 3Laboratory of Genetic Diagnostics, Medical University of Lublin, A. Raclawickie 1, 20-059 Lublin, Poland

**Keywords:** non-Hodgkin lymphoma, children and adolescents, aggressive non-Hodgkin lymphoma

## Abstract

**Simple Summary:**

Non-Hodgkin lymphoma is one of the most frequently occurring hematologic diseases in the world. Current drugs and therapies have improved outcomes for patients with lymphoma, but there is still a need to identify novel medications for treatment-resistant cases. The aim of this review is to gather the latest findings on non-Hodgkin lymphoma in children, including genetic approaches, the application of therapy, the available treatment options, and resistance to medications.

**Abstract:**

One of the most common cancer malignancies is non-Hodgkin lymphoma, whose incidence is nearly 3% of all 36 cancers combined. It is the fourth highest cancer occurrence in children and accounts for 7% of cancers in patients under 20 years of age. Today, the survivability of individuals diagnosed with non-Hodgkin lymphoma varies by about 70%. Chemotherapy, radiation, stem cell transplantation, and immunotherapy have been the main methods of treatment, which have improved outcomes for many oncological patients. However, there is still the need for creation of novel medications for those who are treatment resistant. Additionally, more effective drugs are necessary. This review gathers the latest findings on non-Hodgkin lymphoma treatment options for pediatric patients. Attention will be focused on the most prominent therapies such as monoclonal antibodies, antibody–drug conjugates, chimeric antigen receptor T cell therapy and others.

## 1. Introduction

Non-Hodgkin lymphoma (NHL) is one of the most frequently occurring hematologic disease in the world [1]. In children, NHL comprises four wide categories: lymphoblastic lymphoma, Burkitt lymphoma (BL), diffuse large B-cell lymphoma (DLBCL), and anaplastic large cell lymphoma (Table 1). Among the rarer subtypes are pediatric-type follicular lymphoma (FL) and pediatric marginal zone lymphoma (MZL) [2]. ALK-positive (ALK+) anaplastic large cell lymphoma (ALCL) is the most common subtype of T-NHL in children and represents about 30% of all pediatric lymphomas [3]. The prognosis for NHL patients depends on site of involvement and the pathologic the subtype and stage of the disease; nonetheless, pediatric cases have better outcomes than adults, mainly due to tumor biology [4,5]. Additionally, it is thought that the higher toleration of intensive treatment or distinct pathogenetic mechanisms contribute to better curability in children [4]. Adolescent age or age > 10 years, respectively, has been reported to be associated with inferior outcomes in pediatric trials in patients with mature B-NHL, mainly Burkitt’s lymphoma and DLBCL [6]. The treatment for children diagnosed with NHL includes multiagent systemic chemotherapy with intrathecal administration, surgical intervention mainly used for diagnosing the tumors, while radiation therapy is only for central nervous system involvement, superior mediastinal syndrome, or paraplegias [5].

Overall survival (OS) rates in children, adolescents and young adults diagnosed with NHL increased to 80–90% during the last 30 years, giving the opportunity to investigate the long-term effects of prior chemo- and radiotherapy (RT). Children and adolescent NHL survivors are at significant risk of late mortality from secondary neoplasms, recurrent/progressive disease and chronic health conditions (cardiomyopathy, pneumonia and defects in neurocognitive function), and late morbidity of multiple organ systems and poor health-related quality of life. These risks are similar to other long-term risks of childhood and adolescent acute lymphoblastic leukaemia (ALL), Wilms tumor and Hodgkin lymphoma (HL) survivors [20]. Increasing the dosage of chemotherapeutic or radiation dose not improve the therapeutic response but contributes to the acceleration of side effects development and resistance to therapy [21]. Additionally, 10–30% of pediatric and adolescent patients will relapse; therefore, more effective drugs are necessary [22].

Novel approaches are required to reduce the burden of late morbidity and mortality in childhood and adolescent NHL survivors, and obtain methods to identify at-risk patients who are at significantly increased risk of these complications. Nowadays, we can distinguish several therapeutic substances that work in various ways. These include immunomodulatory drugs, monoclonal antibodies (mAbs), immune checkpoint inhibitors (ICI), antibody–drug conjugates (ADCs) and genetically modified chimeric T cell receptor antigens (CAR) [23]. In addition, research is currently underway on a number of further classes of drugs, the action of which may be based on cross-linking of deoxyribonucleic acid (DNA), inhibition of DNA synthesis, inhibition of the signaling pathway of B-cell receptors, inhibition of proteins regulating apoptosis, or epigenetic modulation [24,25].

More detailed treatment of the different types of NHL in pediatric patients is summarized in Table 2. This review gathers the latest findings on NHL treatment options for pediatric patients.

## 2. New Approaches in Treatment

### 2.1. Monoclonal Antibodies (mAbs) Therapy

Monoclonal antibodies target specific markers on cancer cells by activating the patient’s immune system. In this way, the therapy avoids widespread non-specific cytotoxic effects [31]. Molecular mechanisms that allow mAbs for the destruction of cancer cells are direct cytotoxic actions, immunomodulatory effects and inhibitory effects on promitogenic signaling pathways [32]. There are 50 different mAbs approved by the Food and Drug Administration (FDA). Those directed against antigens expressed on B lymphocytes are ibritumomab tiuxetan, obinutuzumab, ofatumumab, rituximab, brentuximab vedotin (Bv) and alemtuzumab (Table 3) [32]. Rituximab conjugated with chemotherapy is successfully used in most intermediate- and high-risk pediatric mature B-NHL [2]. Additionally, it can also act as a single agent. In this case, studies show that it is effective in high-grade, high-risk, mature B-NHL in children [33].

Some mAbs, such as obinutuzumab and ofatumumab, did not improve outcomes in NHL patients [2]. Nevertheless, obinutuzumab conjugated with chemotherapy was approved for the treatment of FL patients and they can benefit from this therapy [34]. Unfortunately, usage of mAbs for immunotherapy is tied with the appearance of a wide range of side effects, such as anaphylaxis, gastrointestinal symptoms, transient rashes, autoimmunity, cytopenias, toxidermias and pulmonary/cardiac/hepatic/kidney/neurological/embryo–fetal toxicities. Therefore, identification of biomarkers that help with the prediction of a patient’s response is essential for selection of the patients, who are most likely to benefit from mAbs treatment. For example, individuals with positive HER2 gene amplification are classified for trastuzumab therapy or patients with a mutation in codon 12 of the KRAS are unlikely to respond to cetuximab [32]. It is known that the anti-CD19 mAb named B43 conjugated with genistein has entered clinical trials in children with NHL [35].

**Table 3 cancers-14-01569-t003:** mAbs used in NHL therapy in children [32,36].

mAbs name	Generation	Origin	Target	Antigen
rituximab	I	chimeric	B-cells	anti-CD20
obinutzumab	II	humanized		
ofatumumab	I	human		
ublituximab	I	chimeric		
tafasitamab	II	humanized		anti-CD19
inebilizumab	next generation	humanized		
epratuzumab	next generation	humanized		anti-CD22
blinatumomab	next generation	mouse	T cells	anti-CD19 CD3
mosunetuzumab	next generation	humanized		anti-CD20 CD3
glofitamab	next generation	humanized		
odronextamab	next generation	humanized		
epcoritamab	next generation	humanized		

The following generation of mAbs are bispecific antibodies (BiAbs), which are derived from mAbs. They consist of two single-chain variable fragments (scFv) that target tumor-associated antigens. BiAbs engage the cells of immune system to attack indicated tumor cells [37]. Blinatumomab application leads to long-term remission and improved OS rate for patients with relapsed or refractory (R/R) B-NHL. It is approved for treatment both in children and adults [38,39].

### 2.2. Antibody–Drug Conjugates (ADCs)

ADCs comprise of a mAb connected to a small cytotoxic molecule. When attached to the cell-surface antigen of cancer cells, the ADC is internalized; next, the cytotoxin is released, causing cell cycle termination and cell apoptosis. The drug can also kill adjacent cells by “bystander killing” (Table 4) [40]. The design of ADCs depends on the type of antigen, which is present on tumor subtype. Starting with ALCL, whose cells express the CD30 antigen, the Bv is currently under investigation, with its therapeutic potential already proven in adults [41]. CD30 is a member of the tumor necrosis factor receptor superfamily and plays a role in T cell response regulation [42]. Brentuximab consists of an anti-CD30 antibody and anti-microtubule agent monomethyl auristatin E (MMAE), which disrupt mitosis and lead to cell death [43]. Sekimizu et al. involved in their Phase I studies children from 2 to 17 years old with R/R Hodgkin lymphoma or systemic ALCL. Their studies will estimate the safety profile among Japanese children [44]. Meanwhile, results from an ANHL12P1 trial, conducted by the Children Oncology Group (COG) on pediatric patients with ALK+ ALCL, confirmed that Bv combined with chemotherapy outbalanced standard chemotherapy. In these studies, patients below the age of 22 were enrolled (median 12 years old) and achieved a two-year event free survival (EFS) and a two-year OS of 79.1% and 97%, respectively [45]. Another study evaluated Bv alone in the treatment of R/R Hodgkin lymphoma (19 patients) and systemic ALCL (17 patients) in children. In the ALCL group, patients were 2 to 17 years old and 12 of them were ALK+. The safety dosage was estimated at 1.8 mg/kg for phase 2, in which nine patients (53%) of systemic ALCL achieved overall response (seven complete responses (CR) and two partial responses (PR)); for patients with a first relapse of ALCL, six (60%) of them achieved overall responses (four CR and two PR). The adverse effects of this treatment were pyrexia (16/36 patients) and nausea (13/36). Moreover, four patients developed neutropenia [46]. Future perspectives of ADCs usage in pediatric NHLs are sought, for example, for inotuzumab ozogamicin (IO). CD22 is expressed on B-cells, from the pro-b-cell phase to immunoblasts. This antigen plays a great role in the process of proliferation and differentiation of B-cells. CD20 is also present on B-NHL cells [47]. Thus, novel drugs aim at this antigen. IO activates scission of DNA strands in tumor cells with CD20 antigen on their surface [48]. Currently, COG conducts research on treatment with IO in patients diagnosed with B-lymphoblastic lymphoma, or R/R CD22 + B acute lymphoblastic leukemia (NCT02981628). The patients involved in this study are 21 years old or younger [49].

Polatuzumab vedotin, whose clinical trial (NCT02257567) ended in FDA approval for treatment of R/R DLBCL in adults, is the combination of the CD79b antibody and MMAE [50]. Polatuzumab vedotin acts as a microtubule inhibitor and is currently under trial among patients 12–70 years old, suffering from B-NHL or Hodgkin Lymphoma, with previous autologous stem-cell transplantation (auto-SCT) (NCT04491370) [51]. The therapies involving ADCs develop rapidly, and therefore there is a consistent need to seek novel antigens to target, such as JBH492 aiming at CCR7, which is currently under clinical investigation (NCT04240704) [52].

**Table 4 cancers-14-01569-t004:** Comparison of antibodies used in the treatment of pediatric NHLs [2,33,37,38,39,40,41,45,49,52,53,54,55].

Antibody Type	Mechanism of Action	Approved Abs	Abs in Preclinical/Clinical Research
mAbs	activation of apoptosis binding of membrane receptors on the surface of the tumor cell causing inhibition of signal transduction pathway antibody-dependent cellular cytotoxicity complement-dependent cytotoxicities	rituximab (as single-agent therapy or conjugated with chemotherapy)—B-NHL, FL obinutuzumab + chemotherapy—FL	B43 + genistein—NHL epratuzumab galiximab tafasitamab—R/R NHL, FL, DLBCL, MCL MEDI-551—R/R FL, DLBCL
BiAbs	engaging the cells of the immune system to attack malignant cells by targeting the tumor-associated antigen binding to the CD3 antigen on T cells to induce T cell activation and proliferation in an MHC-independent manner	blinatumomab—R/R B-NHL	mosunetuzumab—DLBCL, FL, MCL odronextamab—DLBCL, FL, MCL epcoritamab—DLBCL, FL, MCL plamotamab—DLBCL glofitamab—DLBCL, FL, MCL
ADCs	preferential release of a potent cytotoxic agent at the tumor region, which is caused by proteases or alterations in pH bystander killing	Bv—R/R ALCL polatuzumab vedotin-piiq—DLBCL loncastuximab tesirine-lpyl—DLBCL	Bv—DLBCL Bv + chemotherapy—ALK+ ALCL IO JBH492 Pinatuzumab vedotin—R/R DLBCL, FL Vorsetuzumab mafodotin—R/R NHL Coltuximab Ravtansine (SAR3419) IMGN529 (CD37 ADC)

mAbs, monoclonal antibodies; NHL, non-Hodgkin lymphoma; FL, follicular lymphoma; R/R, relapsed or refractory; DLBCL, diffuse large B-cell lymphoma; MCL, mantle cell lymphoma; BiAbs, bispecific antibodies; ADCs, antibody–drug conjugates; Bv, brentuximab vedotin; ALCL, anaplastic large cell lymphoma; ALK+, anaplastic lymphoma kinase positive; IO, inotuzumab ozogamicin.

### 2.3. Chimeric Antigen Receptor T Cell (CAR-T Cell) Therapy

CAR-T cell therapy uses T lymphocytes, which are engineered with synthetic chimeric antigen receptors (CAR). The CAR-T cell recognizes and then eliminates specific cancer cells, independently of major histocompatibility complex molecules [56]. The creation and mechanism of action of CAR-T cell therapy is shown in Figure 1. The targets of this therapy, among others, are B-cell markers CD19, CD20, and CD22, which are highly expressed in many B-cell malignancies. Unfortunately, such action leads to the elimination of healthy B-cells. For that reason, B-cell maturation antigen (BCMA) is used as an alternative target for CAR-T cell therapy. BCMA is expressed by the plasma cells and the light chain κ/λ of malignant B-cells [53]. The application of second-generation CAR-T cells targeting CD19 with stimulatory domains of CD28/4-1BB has shown significant positive outcomes in the treatment of B-cell lymphomas (FL, PMBCL, DLBCL, mantle cell lymphoma (MCL), and splenic MZL). Furthermore, after using dual specific CD19/CD22-targeted CAR-T cells on a patient with acute B-cell lymphoblastic lymphoma and Li-Fraumeni syndrome, a complete relief of the tumor and negative minimal residual disease (MRD) took place. CAR-T cell therapy is being examined to treat MCL. Application of third-generation CD20-directed CAR-T cells was well tolerated in patients. Additionally, CD20 CAR-T cell treatment resulted in the achievement of CR in BL [53]. Tisagenlecleucel is a second-generation CD19 CAR-T, improving cytokine production, proliferation and CAR-T persistence. It is highly active in children. In a phase II trial, tisagenlecleucel achieved CR for 40% of DLBCL patients. It was estimated that 65% of patients will experience a two-year relapse-free survival [2].

In a recent study of Juan Du, an eight-year-old child suffering from R/R BL showed no clear response after being treated with CD19-specific CAR-T cells. After the attempt to treat the malignancy with CD22-specific CAR-T cells, the disease reoccurred. Subsequently, CD20 CAR-T cell treatment was applied, and that action resulted in the achievement of CR. What is more, CAR-T cell therapy targeting CD23 and the tyrosine kinase-like orphan receptor brought about promising results in the improvement of R/R NHL treatment in children [57]. One of the favorable qualities of using CAR-T for pediatric NHL is that the infusions require minimal additional chemotherapy before administration [58].

### 2.4. DNA Methyltransferase (DNMT) Inhibitors

DNMT inhibitors belong to a family of enzymes that catalyze the methylation of DNA [59]. There are three major types of DNMT in mammals: DNMT1, DNMT3a, and DNMT3b [60]. DNMTs have a huge impact on gene regulation process and they are a potential predictive biomarker of genetic disorders and diseases [59]. Methylation is a post-translational modification, whose biological role is to preserve DNA transcriptionally on hold. Such action results in gene silencing. DNMT inhibitors cause cell cycle and growth arrest, differentiation and apoptosis. The molecular mechanisms by which DNMT inhibitors induce anti-cancer effects are partially mediated by the hypomethylation of DNA, which at higher concentrations may lead to cytotoxic effects (Figure 2) [21].

The conclusions of Han Weidong’s study on children under 16 years of age inform us that decitabine-primed CAR-T cells can recognize and kill the CD19 negative malignant cells, which leads to death of lymphoma tumor cells. Decitabine increases tumor antigens and human leukocyte antigen expression, enhances antigen processing, promotes T cell infiltration and boosts effector T cell function; therefore, it can be used in DLBCL, high grade B-cell lymphoma and other aggressive B-cell lymphomas in pediatric patients [61]. Decitabine is also thought to be effective in R/R T-lymphoblastic lymphoma treatment in people who are over 14 years old [62].

### 2.5. Histone Deacetylase Inhibitors (HDACIs)

For many years, the modulation of epigenetic mechanisms was the subject of trials and studies [63]. The HDACIs influence the degree of acetylation in non-histone proteins, which results in alteration of certain proteins functions, thereby affecting the cell cycle [64,65]. The role of HDACIs in lymphoma treatment, as well as their detailed mechanism of action, was extensively described in a recent review by Chen et al. [66]. Simplified mechanism of action is presented on the Figure 3.

At the time of this article’s publication, there are two ongoing pediatric clinical trials involving vorinostat in NHL. They cover combination with chemotherapy before donor stem cell transplant (NCT04220008) and a potential graft-versus-host disease (GVHD) incidence reducing drug (NCT03842696) [67,68]. Promising results were brought by van Tilburg et al., who managed to estimate a safe dosage of vorinostat in several types of pediatric malignancies [69].

In the case of panobinostat, the results of only one clinical trial (NCT01321346) were published by Goldberg et al. Among 22 pediatric patients, only one was diagnosed with NHL. The researchers observed several adverse effects, predominantly regarding the gastrointestinal tract. Moreover, the clinical activity of panobinostat was unsatisfactory, and now more efficient therapies are available [70].

The following examples of HDACIs were registered in trials in patients at least 16 years old. With a positive outcome of these studies, we may assume the age of enrollment to these trials will be lowered in the future.

Chidamide is being tested in R/R peripheral T cell lymphoma with encouraging results and a satisfying safety profile [71]. Moreover, the NCT03151876 brought the results of a favorable combination of chidamide with cladribine, gemcitabine and busulfan. Adults patients with a high risk or R/R NHL reached the OS of 86.1% [72]. Moreover, Wang et al. describe treatment with chidamide combined with various chemotherapy protocols. Their patients were diagnosed with hepatosplenic T cell lymphoma. Due to limited data, the conclusions are not clear. However, 7 out of 14 patients responded to therapy, with three patients achieving CR [73].

HDACIs are undeniably under rapid development. Certainly, we will observe more clinical trials of either HDACIs as single agents or in combined therapies with novel drugs [74].

### 2.6. Immune Checkpoint Inhibitors (ICIs)

ICIs have become a great milestone as a cancer cure. Pediatric malignancies, such as leukemias and solid tumors, also benefit from ICIs, with their expansion in our sight [75,76]. Despite limited data, there are a few clinical trials involving NHL treatment in children.

As for nivolumab, which is a fully human IgG4 mAb targeting the PD-1 receptor (Figure 4), there are very limited data in the field of this review [77]. The first results of the NCT02304458 clinical trials were collected with the conclusions of safety and tolerance of nivolumab in a group of pediatric patients [78]. NHL constituted only 10 of the 72 patients of “Group B”, with a median age of 15. However, only eight NHL cases were tested for PD1 expression, of whom seven were positive. Despite this fact, only one patient responded to PD-1 blocking therapy, which resulted in the termination of testing nivolumab as a single agent for this group [79]. However, there may be a subgroup of patients that can benefit from anti-PD-1 receptor therapy. A case of a 17-year-old patient, with ALK+ ALCL, received nivolumab as the third line therapy, obtaining a CR [80]. With regard to the patient’s age, we are aware that further studies are necessary for pediatric patients; nonetheless, with the promising results in the adults’ group, we hope to achieve a positive outcome with PD-1 blockade [81,82].

In terms of other checkpoint inhibitors, the fully human IgG1 PD-1L antibody, atezolizumab, has become the object of clinical trials [83]. However, the study (NCT02541604) was terminated, and in NHL, only three patients were enrolled (3% of all participants), achieving poor results with one partial response and one progression of the disease. Nevertheless, atezolizumab was proved to be safe in the pediatric group of patients. Due to limited data, the results of the treatment were highly reduced.

There are several active clinical trials, which will determine the usage of the mentioned ICIs, along with others, such as anti-CTLA-4. The National Cancer Institute is conducting studies (NCT02304458) on combined nivolumab with ipilimumab (which is the anti-CTLA-4 antibody) in patients with R/R NHL [84].

### 2.7. Enhancer of Zeste Homolog 2 (EZH2) Inhibitors

EZH2 is a histone methyltransferase, which comes from the gene family containing epigenetic regulators that inhibit transcription [85]. EZH2 alters gene expression by trimethylation of Lys-27 in histone 3 (H3K27me3) [86,87]. Thus, EZH2 can act as a regulator of cell cycle progression, and its mutations will lead to abnormal histone methylation profiles, resulting in oncogenic transformation [88]. In B-cells, EZH2 is highly expressed in the germinal center (GC), where maturing B-cells undergo somatic mutations to form a series of sublines. Due to the fact that EZH2 is involved in the mechanisms of checkpoints, it influences the maintenance of an orderly process in the GC, thus controlling the transition of lymphocytes from pro-B to pre-B [89].

When EZH2 mutation occurs, the suppression of GC output genes and checkpoints persists, which in turn leads to hyperplasia [90]. Until now, EZH2 somatic mutations associated with the acquisition of function were most often detected in DLBCL and in GC-derived FL [91,92]. In this context, efforts to find inhibitors of EZH2 have become understandable.

Since 2012, several specific EZH2 inhibitors have been investigated. Their task was to inhibit H3K27 methylation, reactivate silenced PRC2 target genes, or inhibit the survival of B-cell lymphoma cells (GC-derived and containing the EZH2 activating mutation) [93]. These selective compounds included: El1, UNC1999, OR-S1 and OR-S2 inhibitors, EPZ0005687, and GSK126, CPI-360, CPI-169, DS-3201 (Walemetostat), and EPZ-6438 (tazemetostat) [94,95,96,97,98]. The last of them, tazemetostat, as a compound with high affinity to mutant forms of EZH2, is currently being evaluated in clinical trials in the treatment of pediatric patients with R/R NHL [99,100,101,102].

EZH2 inhibitors have the potential to be useful in the treatment of lymphomas, but there is still insufficient research to draw specific conclusions. It is possible that in the future it will be possible to use compounds from this group in combination therapy, e.g., with chemotherapy, which would result in improved patient outcomes. However, for this to happen, a lot of research has yet to be conducted.

### 2.8. Isocitrate Dehydrogenase (IDH) Inhibitors

IDH is an enzyme that catalyzes the conversion of isocitrate to α-ketoglutarate (αKG) [103]. Recently, more and more has been said about mutations related to these dehydrogenases, as they are detected in many cancers. IDH1/2 mutations may involve both loss of enzyme function and an increase in its activity. They lead to the disturbance of the oxidative decarboxylation of isocitrate to α-KG, additionally giving new enzymatic activity, resulting in the reduction of α-KG to the oncometabolite D-2-hydroxyglutarate (D-2HG) [104,105]. There are reports of IDH mutations in angioimmunoblastic lymphomas, classified as NHLs [104]. Due to the prevalence of mutant IDH in tumors, efforts were made to develop inhibitors of this dehydrogenase. Currently, the group of drugs that are IDH inhibitors includes enasidenib, ivosidenib (AG-120), AG-881, FT2102, and IDH-305 [103].

Enasidenib and ivosidenib are orally selective inhibitors of mutant IDH, with enasidenib blocking IDH2 and ivosidenib blocking IDH1. Both of these drugs have already been approved by the FDA for the treatment of acute myeloid leukemia (AML) [106,107,108].

Further clinical trials are currently underway to test these compounds in other diseases as well. One of these studies, on ivosidenib, is currently being conducted in the age group 12 months to 21 years. It focuses on patients with R/R NHL, advanced solid tumors, or histiocytic disorders that have IDH1 genetic alterations [109].

### 2.9. The B2 Cell Lymphoma Protein (BCL-2) Inhibitors

Expression of anti-apoptotic proteins from the BCL-2 family can be disrupted by several different mechanisms, including gene amplification, chromosomal translocation, increased gene transcription, or altered post-translational processing [105]. In the case of FL and DLBCL, it has been shown that increased BCL-2 expression may result from the t(14; 18) translocation of this gene [110,111,112]. The BCL-2 gene is then inserted proximate to the immunoglobulin heavy chain (IgH) gene enhancer [113]. The consequence of this is activated transcription of the BCL-2 gene and, thus, inhibited cell apoptosis. It is worth mentioning that in NHL subtypes, there may also be changes leading to overexpression of BCL-2, regardless of t(14;18). This can occur, for example, by the aforementioned amplification of this gene [114]. From a therapeutic point of view, it therefore appears beneficial to target regulators of apoptosis in cancer patients with mutations within the BCL-2. This could ultimately enhance the apoptotic response and thus translate into improved clinical outcomes.

Currently, several compounds that are inhibitors of BCL-2 can be distinguished. One of them, ABT-263 (navitoclax), initially showed good efficacy. However, its use has been limited due to its main toxicity: thrombocytopenia. Despite this, research can still be found in the context of NHL and navatoclax. In one study, although each patient had at least one treatment-related adverse event, the safety profile of navitoclax in R/R FL was assessed as acceptable [115]. In turn, ABT-199 (venetoclax/GDC-0199) shows a high binding affinity to BCL-2, where it exerts its clinical effect without the simultaneous thrombocytopenia [116,117]. Recent studies suggest that venetoclax has good clinical outcomes. An example may be the study by Davids et al., who decided to evaluate the effect of venetoclax on a group of patients with R/R NHL. In their studies (although the results differed by NHL subtype), they achieved overall responses rates (ORRs) in 75% of patients with MCL and 38% with FL. Although only 5/29 FL patients and 6/28 MCL patients achieved CR, all of them had a median duration of response greater than 30 months [118,119]. For this reason, many studies on its effectiveness are currently underway, including several involving a group of children with NHL [120,121]. These studies focus not only on the safety assessment and pharmacokinetics of venetoclax in children with R/R malignancies (including NHL), but also investigate the effect of combination therapy (venetoclax + chemotherapy) on the treatment of R/R lymphoblastic lymphoma (LL) [122]. It is worth mentioning that a study evaluating venetoclax + navitoclax + chemotherapy in children and adults with R/R acute lymphoblastic leukemia or LL has recently ended. In this study, the CR rate was 60%, and 28% of patients crossed over to either transplantation or CAR-T therapy [123].

In the context of the future, VOB560 (65487) and MIK665 (S64315) may also be significant. Both of these compounds were designed to potently and selectively block BCL-2 and MCL-1, respectively. Their combination in preclinical studies has shown strong anti-cancer properties [124]. 

Undoubtedly, the discovery of inhibitors of the BCL-2 family opened a new path in the targeted therapy of many cancers. There is the need for new clinical trials comparing the effectiveness of these substances in similar and the same diseases. Their toxicity profile is also extremely important. As already proven, the side effect of individual BCL-2 inhibitors is not only the abovementioned thrombocytopenia. In addition, with venetoclax, diarrhea, upper respiratory tract infections, neutropenia and tumor lysis syndrome may occur [125,126,127]. However, for the other substances mentioned, we have to wait for the completion of ongoing studies [116]. To be able to efficiently operate these agents, we must also be aware of the potential resistance of cancer cells to BCL-2 inhibitors; this has already been observed in the case of navitoclax. Although samples of CLL patients initially responded to treatment, there was a 1000-fold increase in resistance to navitoclax over a longer period of time [128]. A similar situation occurred with the use of venetoclax [129,130]. A solution to this situation may be the synergism of BCL-2 and MCL-1 inhibitors described above.

### 2.10. Anaplastic Lymphoma Kinase (ALK) Inhibitors

Approximately 90% of pediatric ALK+ ALCL is due to t(2;5) (p23;q35) chromosomal translocation, which entails the emergence of oncogenic fusion protein nucleophosmin activating several proliferation and survival pathways [131]. The reason why ALK is one of the targets of personalized therapies is that ALK-transformed cells are dependent on ALK tyrosine kinase activity in terms of survival and proliferation. Because ALK expression is limited in non-tumoral cells, its blockage is catastrophic for cancer cells but irrelevant for normal tissues (Figure 5) [132].

Crizotinib is a first-generation ALK inhibitor [133]. Based on the tolerability, toxicity and activity of single agent crizotinib, it was demonstrated that the discussed molecule synergizes with chemotherapy [134]. Crizotinib in combination with multiagent chemotherapy is being evaluated in an ongoing randomized NCT01979536 trial in childhood ALCL [27]. In an NCT02034981 trial investigating crizotinib in pediatric ALK+ ALCL patients, 1/3 of the examined children progressed in the first three months of treatment [135]. On the other hand, the secondary outcomes of NCT00939770 trial showed that 70% of pediatric patients with R/R ALK+ ALCL treated with crizotinib achieved CR or PR within eight years [136]. Recently, crizotinib was approved by the FDA for R/R ALK+ ALCL in treatment for patients aged over one year, as it was well tolerated and led to a CR in 67–83% of children with R/R ALCL [28].

Ceritinib is a second-generation ALK inhibitor that binds to ALK with a higher affinity than crizotinib [133], but its use is associated with significant toxicities, limiting its application to children and adolescents [137]. Still, in a phase I NCT01742286 trial, where single-agent ceritinib in pediatric patients with ALK+ malignancies was examined, two out of two children with ALK+ ALCL achieved CR [29].

### 2.11. Ibrutinib/Bruton’s Tyrosine Kinase (BTK) Inhibitor

BTK is a kinase with a protein structure involved in the regulation of B-cell signaling [138]. As one of five members, it belongs to the TEC family of non-receptor tyrosine kinases [139]. BTK is expressed in B-cells, bone marrow cells, mast cells and platelets [140]. This kinase, as a key component of the B-cell antigen receptor (BCR) signaling cascade, is involved in all aspects of B-cell development [141]. However, if its action is unregulated, it leads to uncontrolled proliferation, differentiation, consequent survival of B-cells and, hence, the development of cancer [138]. So far, BTK expression has been described in B-cell leukemias and lymphomas. Its involvement in the regulation of FcγR signaling and in mast cell degranulation after FcεR activation has been observed. These features make it an attractive target for the treatment of both B-cell cancers and autoimmune diseases, leading to the development of BTK inhibitors [140].

Several generations can be distinguished among these drugs. The first-generation BTK inhibitors include ibrutinib, which is administered orally. It binds irreversibly to a cysteine residue (C481) in the active site of BTK, thereby inhibiting B-cell receptor signaling (through decreased activation of extracellular signal regulated kinase, PLCγ2, and the NF-κB signaling pathway) [142,143]. As a result, it reduces cell growth, proliferation, survival, adhesion, and migration [141,144].

Studies to date have shown that ibrutinib is well tolerated in many B-cell tumors, including DLBCL [145,146]. However, most of them still relate to adults. As for children, a study evaluating the safety and efficacy of ibrutinib in a group of patients from 1 to 30 years with R/R NHL was completed at the end of last year. However, its results have yet to be published (NCT02703272) [147,148]. Nevertheless, we already know that it is not a drug without its drawbacks. This is due to the fact that it can cause common side effects such as diarrhea, nausea, and shortness of breath. Moreover, there have also been cases of thrombocytopenia, bleeding, and atrial fibrillation [149]. For this reason, the second-generation of BTK inhibitors has been developed, including acalabrutinib (ACP-196), zanubrutinib (BGB-3111), and tirabrutinib (ONO/GS-4059) [150,151,152,153,154]. These inhibitors, although similar to ibrutinib, bind covalently to the C481 residue in BTK, exhibiting higher target selectivity. It is possible that this could reduce the frequency of side effects, including the cardiological ones. Currently, acalabrutinib and zanubrutinib have already received FDA approval for treatment [153,155,156,157,158].

It is worth mentioning that, as in the case of other drugs, the possibility of combining BTK inhibitors with, e.g., BCL2 inhibitors, is being considered. In one such study, Constantine et al. proved that the combination of ibrutinib + venetoclax was highly effective in patients with MCL. They received an OS rate of 79% at 12 months and 74% at 18 months [159]. In turn, in the study by Amos Burke et al. evaluating children with R/R mature B-NHL, 57.1% of patients who received ≥ 1 dose of ibrutinib responded. In contrast, 72.7% of patients who received ibrutinib with modified RICE (rituximab plus ifosfamide, carboplatin, and etoposide) responded. This proves the validity of further research of this type in the future [160].

### 2.12. Bortezomib, Inhibitor of Proteasome

Bortezomib (PS-341) is a peptide aldehyde derivative. It is the first potent, selective, and reversible inhibitor of the 26S proteasome in its class (Figure 6) [161]. 

Its role is to inhibits the ubiquitin–proteasome pathway by binding directly to the active sites of the proteasome, which in turn disrupts targeted protein proteolysis [161,162]. The consequence of this is the deregulation of homeostatic mechanisms at the cellular level, leading to apoptosis and inhibition of cell cycle progression, angiogenesis, cell adhesion, and proliferation. It is worth mentioning that bortezomib has also been shown to induce apoptosis in cells overexpressing BCL2 [161,163]. Initially, its development was to be used in inflammation and cachexia. However, this changed in the late 1990s, when its anti-cancer activity was established [162,164,165]. Now, it is approved by the FDA for the treatment of MCL. This decision was based on a phase III study, in which a higher CR rate was obtained in the group of patients with the discussed drug [166,167]. Bortezomib has been found to be effective in other NHLs as well. In the study by Ruan et al., combining this drug with R-CHOP in the DLBCL cohort resulted in a CR/CR unconfirmed (CRu) percentage of 86%. In addition, the study authors concluded that such treatment was safe and could improve outcomes, especially for patients with non-germinal center DLBCL [168,169]. In the case of bortezomib, there are also several side effects that it can cause. The most important of these is peripheral neuropathy, which is usually sensory, distal, and symmetrical, and affects the feet more often than the hands. In most cases it is reversible on stopping treatment [170]. In addition, its risk has been shown to be lower with subcutaneous bortezomib compared to intravenous administration [161]. Other side effects that may be mentioned are gastrointestinal toxicity (such as diarrhea or constipation), fatigue, neutropenia, and thrombocytopenia. Shingles may also reactivate during bortezomib treatment, and antiviral prophylaxis is usually recommended [170]. Of note is isolated cutaneous vasculitis caused by bortezomib, which does not always lead to treatment discontinuation. It has been shown that it may herald a better clinical response in some patients with B-NHL [171].

Although most studies on bortezomib in NHL patients concern adults, for several years there have also been studies in which the target group are children. One of them was conducted by Horton et al. and related to the use of combination therapy of bortezomib with ifosfamide/vinorelbine (IVB) in pediatric NHL patients. Although few patients achieved the primary goal (which was complete anatomical response after two IVB cycles), the CR after two cycles was 83% [172,173]. Recently, a study evaluating the effect of bortezomib on combination chemotherapy in patients with R/R NHL has been completed. The BICE group (bortezomib, ifosfamide, carboplatin, and etoposide) achieved a slightly higher OR rate (70%) compared to the group without bortezomib (60%). However, to be able to talk about the effectiveness of bortezomib in the treatment of children with NHL, much more research is needed [174].

### 2.13. Temsirolimus

Temsirolimus is one of the mammalian targets of the rapamycin (mTOR) inhibitor, which, as a derivative of the sirolimus ester, exhibits antibacterial, immunosuppressive, and antitumor properties [122,175]. The main function of the mTOR protein kinase is the regulation of growth and proliferation, as well as the translation and transcription processes [176,177].

Currently, temsirolimus is approved in the European Union for the treatment of R/R MCL, but not in the USA [122,178,179]. The discrepancy in the approval of temsirolimus in the NHL is likely due to the fact that although it has been shown to be effective, the response rate is usually well below 50%. In addition, the median disease-free survival is often a few months or less [180,181,182].

For this reason, combinations of temsirolimus with other agents, both cytotoxic drugs and other targeted inhibitors, are currently at the forefront of NHL treatment [180]. One such study was conducted by Fenske et al. and related to the combination of temsirolimus with bortezomib in R/R B-NHL patients. As the treatment did not cause any unexpected toxic effects in any of the patients, this combination was assessed as safe. The OR rate was 31% and the median progression-free survival (PFS), although different by the B-NHL subtypes, in patients with MCL was 16.5 months. Thus, the observed results justify further research in this direction [122]. In turn, the study by Hess et al. evaluating the combination of bendamustine + rituximab + temsirolimus (BeRT) in R/R MCL and FL also achieved promising efficacy with an acceptable toxicity [183]. As it turns out, temsirolimus can be effectively used in other NHL subtypes. A study by Witzens-Harig et al. confirmed that it can be safely added to the rituximab regimen in combination with dexamethasone, cytarabine, and cisplatinum (R-DHAP). The ORR in this study was 66%, while the two-year overall PFS and OS were 53% and 59% [184]. As can be seen, many studies involving temsirolimus in NHL have already been conducted in adults, and many more are still in clinical trials. In the case of children there are fewer of them, but we can also distinguish a few. These include studies evaluating the side effects and the best effective dose when combining temsirolimus with dexamethasone, mitoxantrone hydrochloride, vincristine sulfate, and pegaspargase in young NHL patients, as well as studies evaluating the combination of temsirolimus with etoposide [185,186].

As can be seen, despite the approval of temsirolimus for treatment in MCL, there is still a space for improvement in its effectiveness in this and other types of NHL. It is possible to achieve this by developing better biomarkers, which would allow for better stratification of the patient prior to drug administration. Moreover, such biomarkers could help to elucidate the mechanisms of resistance and to develop new therapeutic combinations, which would certainly improve the effectiveness of treatment [180].

## 3. Conclusions

Our knowledge of the refractory NHL’s treatment has grown significantly. Novel drugs trial results give hope to those who suffer from the mentioned hematologic dis-eases and make up a large percentage of people.

There are many approved drugs for NHL therapy, e.g., temsirolimus for MCL or bilinatumomab for R/R B-NHL in children and adults. The application of second-generation CD19 CAR-T cells has shown significant positive outcomes in the treatment of FL, PMBCL, DLBCL, MCL, and splenic MZL. Moreover, CD20 CAR-T cell treatment resulted in CR in BL.

EZH2 inhibitors in combination with chemotherapy will probably improve patients’ outcomes, if examined more closely. When the effective doses have been improved and the side effects suppressed, BCL-2 inhibitors will be a promising class of drugs to fight NHL. Additionally, combination of ibrutinib and venetoclax is considered highly effective in treating MCL. Lastly, chidamide indicated encouraging results in its effectiveness and safety profile in T cell lymphoma treatment; for that reason, we can expect many clinical trials validating this method.

There is a necessity to develop better biomarkers, which could help with elucidation of resistance mechanisms and stratification of the patient before administrating the drug.

It is likely that the majority of the compounds described in this review will be used in widespread NHL therapies; however, much more research needs to be conducted and many years have to pass to establish personalized treatment options. As always, future pediatric treatment regimens will arrive after clinical trials take place in adults. Thus, we expect rapid progress in this branch of medicine. All figures presented in this article have been created with Biorender.com.

## Figures and Tables

**Figure 1 cancers-14-01569-f001:**
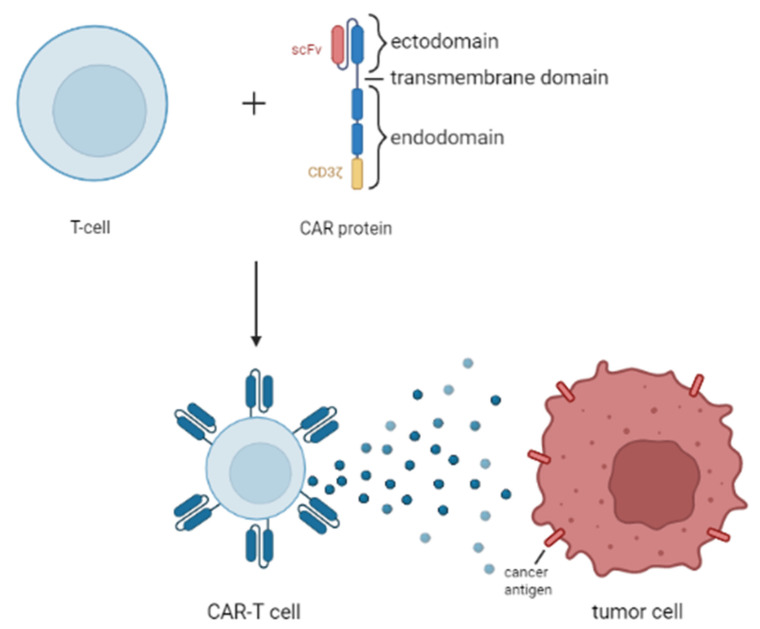
Structure of CAR-T cells and their antitumor function.

**Figure 2 cancers-14-01569-f002:**
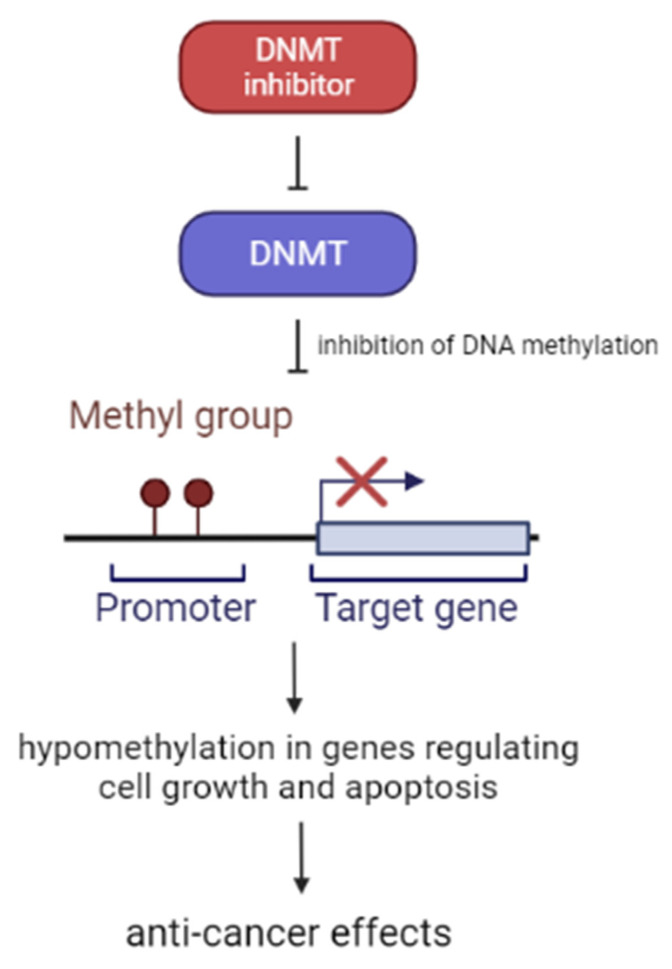
Effect of DMNT inhibitors application.

**Figure 3 cancers-14-01569-f003:**
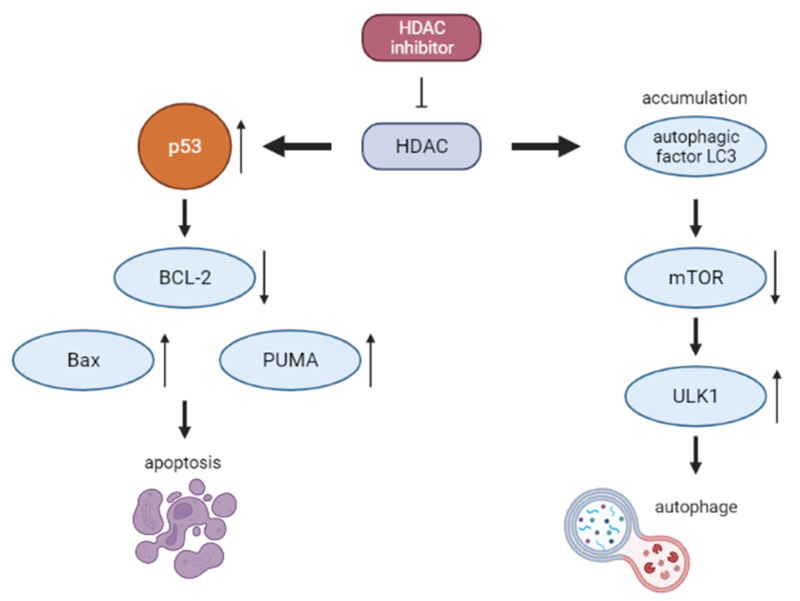
Mechanism of action HDACI.

**Figure 4 cancers-14-01569-f004:**
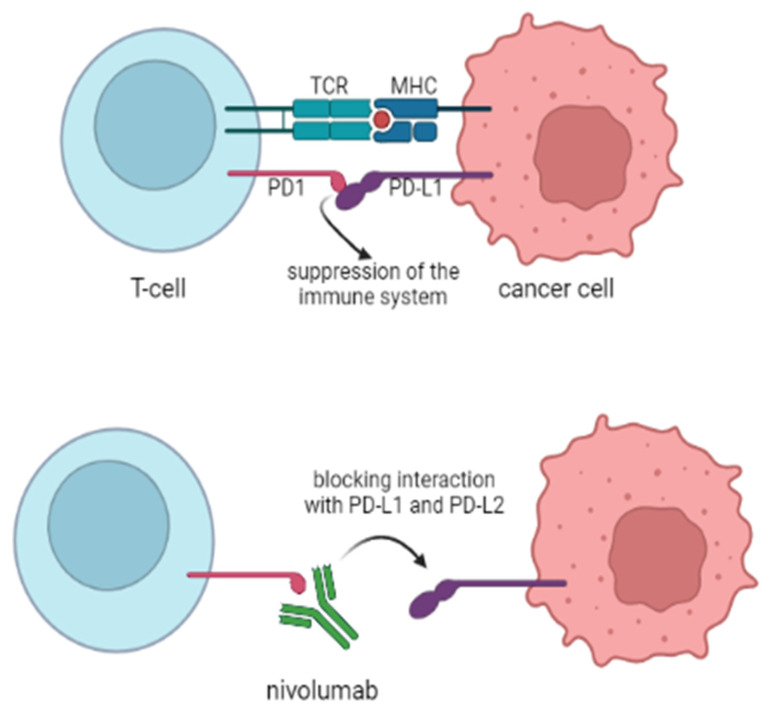
Construction of the CAR-T cell.

**Figure 5 cancers-14-01569-f005:**
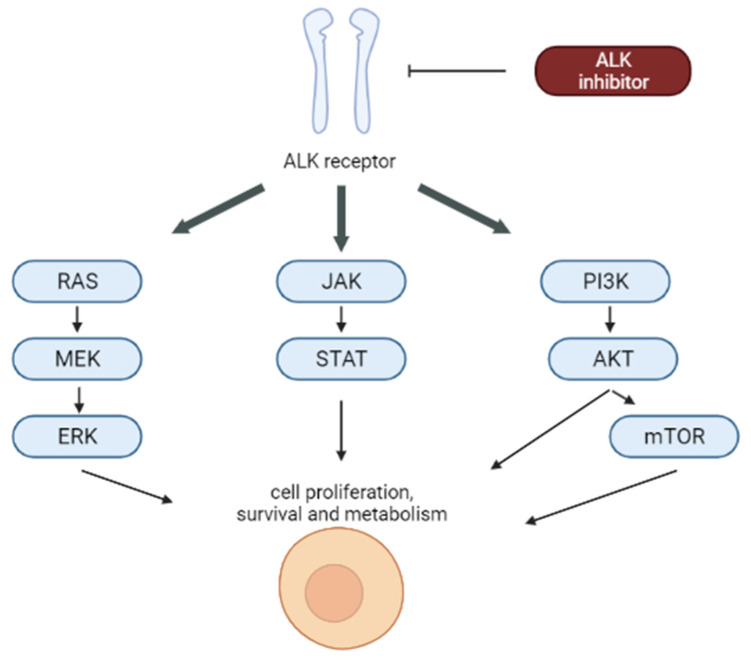
Signalling pathways blocked by ALK inhibitors.

**Figure 6 cancers-14-01569-f006:**
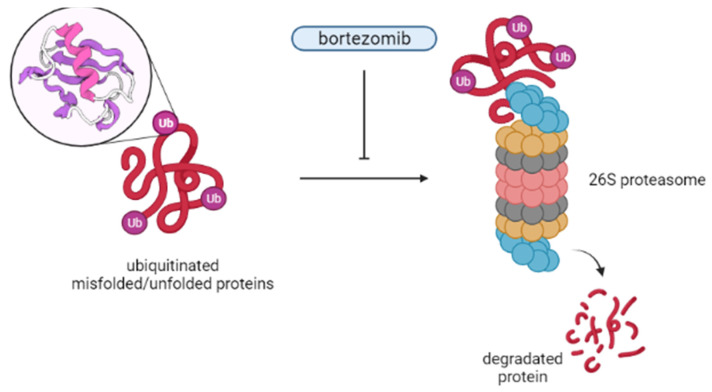
Bortezomib mechanism of action.

**Table 1 cancers-14-01569-t001:** Characteristic of main pediatric NHL types [3,5,7,8,9,10,11,12,13,14,15,16,17,18,19].

NHL Type	Frequency of Occurrence in Pediatric NHLs	NHL Subtypes	Clinical Features	Genetic Rearrangements and Prognosis	
B-NHL	86%	DLBCL BL BLL PMBCL pediatric type FLpediatric nodal MZL	generally localized lesionsabdominal tumor nasopharyngeal tumor jaw bone tumor solid tumor syndrome	DLBCL	
				*MYC*—8q25 rearrangements	poor prognosis
				t(14;18)(q32;q21) *IGH::BCL2*	poor prognosis
				*BCL6*—3q27 rearrangements	poor prognosis
				t(6;14)(p25;q32) *IGH::IRF4*	favorable outcomes
				translocations *MYC/BCL2, MYC/BCL6, MYC/BCL2/BCL6*	poor prognosis
				BL	
				*c-MYC* translocations: t(8;14)(q24;q32) *IGH*::*MYC*, t(8;22)(q24.1;q11.2*) IGL*::*MYC*, t(2;8)(p12;q24.1) *IGK*::*MYC*	poor prognosis
				del(13q14.3) or del(13q34)	poor prognosis
				ID3-TCF3-CCND3 pathway mutations	no correlation to the outcome
				BLL	
				11q aberration with proximal gains and telomeric losses	favorable outcomes
				FL	
				In pediatric FL, there rarely occurs t(14;18), which is typical for FL; lack of this translocation correlates with excellent outcomes in pediatric FL	
LBL-T/B	1–4 years old 40% 15–19 years old 20%	LBL-T (75%) LBL-B	mediastinal tumor pleural effusion respiratory failure not likely to involve the CNS at diagnosis or relapse relapse into the marrow	LBL-T	
				Chromosomal abnormalities including TCR genes, e.g., translocations in *TAL1, LMO2, LYL1, HOXA9, TLX1, TLX3*	unknown
				t(7;14)(p15;q32) *HOXA::TCL1A*	unknown
				Notch1 mutations +/− FBXW7 mutations	favorable outcomes
				LOH6q16	poor prognosis
				ABD	poor prognosis
				*PTEN* mutations	poor prognosis, unless presence of notch1 or absence of LOH6q
				*PHF6* mutations	favorable outcomes
				*NRAS/KRAS* mutations	no correlation to the outcome
ALCL	Median around 16 years −10%	ALCL extra-nodal NK/T cell lymphoma T cell hepatosplenic lymphoma subcutaneous panniculitis like T cell lymphoma	mediastinal tumor	t(2;5)(p23;q35) *NPM1::ALK*	unknown; although t(2;5) is found in aggressive high grade tumors, a 80% 5-yr survival seems to be associated with this anomaly
			tumors in the digestive tract		
			peripheral, mediastinal, or abdominal lymphadenopathy		
			hepatosplenomegaly		
			skin changes		
			changes in the lung parenchyma		
			extra-nodal lesions (brain, marrow, bones, liver, spleen)		
			associated hemophagocytic lymphohistocytosis		

NHL, non-Hodgkin lymphoma; BL, Burkitt lymphoma; DLBCL, diffuse large B-cell lymphoma; PMBCL, primary mediastinal B-cell lymphoma; FL, follicular lymphoma; MZL, marginal zone lymphoma; LBL-T, T cell lymphoblastic lymphoma; LBL-B, B-cell lymphoblastic lymphoma; CNS, central nervous system; ALCL, anaplastic large cell lymphoma; ID3, inhibitor of DNA binding 3; BLL, Burkitt-like lymphoma; TCF3, transcription factor 3; ALK, anaplastic lymphoma kinase gene; NPM, nucleophosmin gene; ABD, absence of biallelic deletion of the T cell Receptor Gamma(TRG) locus.

**Table 2 cancers-14-01569-t002:** Treatment of different types of NHL in children [3,7,10,26,27,28,29,30].

NHL Type	Classical Treatment	Treatment after Lack of Response to Classical Treatment or Relapse	Novel Treatment Options
B-NHL	rituximab prednisone vincristine methotrexate doxorubicin arabinoside cyclophosphamide etoposide	ibrutinib mega chemotherapy + allo-HSCT	mAbs (obinutuzumab) ADCs (inotuzumab) CAR-T cell therapy ICIs (pembrolizumab) pathway inhibitors (buparlisib, ibrutinib)
LBL-T/B	multidrug chemotherapy	chemotherapy with nelarabine, cyclophosphamide and etoposide mega chemotherapy + auto/allo-HSCT	ruxolitinib tyrosine-serotonin kinase inhibitors gamma secretase inhibitors
ALCL	methotrexate combination of cyclophosphamide, doxorubicin, vincristine, corticosteroids, ifosfamide and etoposide tumor removal surgery	allo-HSCT vinblastine re-induction salvage chemotherapy + auto-SCT re-induction salvage chemotherapy + alloSCT	mAbs Bv kinase inhibitors (ceretynib)ICIs (nivolumab) signaling pathway inhibitors (ruxolitinib) anaplastic lymphoma kinase inhibitors (crizotinib, alectinib, ceritinib)
ALK+ALCL	doxorubicin-containing polychemotherapy, typically CHOP 3-week induction therapy (vincristine, prednisone, cyclophosphamide, daunomycin, asparaginase) followed by a 3-week consolidation period (vincristine, prednisone, etoposide, 6-thioguanine, cytarabine, asparaginase, methotrexate), subsequently 6 courses of maintenance chemotherapy (cyclophosphamide, 6-thioguanine, vincristine, prednisone, asparaginase, methotrexate, etoposide, cytarabine) at 7-week intervals HDC/ASCT	HDC/ASCT allo-SCT	crizotinib crizotinib + multiagent chemotherapy ceritinib Bv

NHL, non-Hodgkin lymphoma; allo-HSCT, allogenic hematopoietic stem-cell transplantation; mAbs, monoclonal antibodies; ADCs, antibody–drug conjugates; CAR-T, chimeric antigen receptor T; ICIs, immune checkpoint inhibitors; LBL-T/B, T/B-cell lymphoblastic lymphoma; ALCL, anaplastic large cell lymphoma; auto-SCT, autologous stem-cell transplantation; all-SCT, allogenic stem cell transplantation; Bv, brentuximab vedotin; CHOP, cyclophosphamide, doxorubicin, vincristine, prednisone; ALK+, anaplastic lymphoma kinase positive; HDC/ASCT, high-dose chemotherapy supported by autologous stem cell transplantation.

## Data Availability

No new data were created or analyzed in this study. Data sharing is not applicable to this article.

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
