# Peer review of "The New Treatment Methods for Non-Hodgkin Lymphoma in Pediatric Patients"

_cancers, 2022, doi:10.3390/cancers14061569_

Round 1

Reviewer 1 Report

The review written by Derebas and colleagues tries to gives an overview on contemporary treatment strategies for children and adolescents with non-Hodgkin lymphoma (NHL) with a special focus on novel treatment concepts and targeted therapies based on immunologically and genetically defined subtypes.

The manuscript covers relevant aspects of contemporary treatment for pediatric NHL, however, it has major limitations:

For a better reading, I first suggest to shorten and re-work the text substantially.

The introduction section contains detailed information on NHL in general, which most readers with a pediatric hemato-oncological background are usually familiar with. I suggest to shorten this part considerably. In line 40 they write “Boys develop HNL three times less often than girls, but in adolescents …, the disease of girls prevails (Ref 6)”. I cannot check reference 6, as this is an e-book in polish language, but I am sure that this statement is not true. In general, a male preponderance is reported in the literature for pediatric NHL (Burkhart, Brit J Haematol 2005 and Leukemia 2011; Reedijk, Eur J Cancer 2022).

In the second section (NHL treatment), they give an overview on the current treatment strategies for NHL and discuss well-known disadvantages of chemotherapy, radiotherapy and stem cell transplantation. Next, the rationale for alternative drugs or targeted therapies according to the different NHL subgroups is explained. As overview, Table 2 is provided. Table 2 contains a few misspellings such as “rituksymab” and “ibrutynib”– this needs to be corrected accordingly.

In the upcoming section, different therapies are addressed. The sub-sections for monoclonal antibodies and antibody-drug conjugates are well written and straight to the point. The abbreviation Bv in line 143 needs to be defined (Brentuximab vedotin).

The next sub-section on CAR-T cell therapy (starting at line 212) is much too long. I think, that explaining detailed signalling mechanisms and the evolution of different CAR-T generations is beyond the scope of this review. Instead, I suggest focusing on CAR-T cell targets relevant for pediatric NHL subtypes and to discuss experimental data and results available from clinical trials.

Furthermore, the sub-sections on EZH2 and IDH inhibitors both should be shortened substantially.

In the next sub-section, starting at line 428, the authors explain the role of BCL-2 associated proteins in apoptotic signalling very much in detail, which is not necessary and again beyond the scope of this review. The authors should try to summarize, what information on BCL-2 is essential with respect to the treatment of NHL subtypes and which inhibitors are currently available or tested in clinical trials. In addition, they must discuss their limitations and provide clinical data, as done nicely for venetoclax. However, the evolution of these compounds from ABT-373 to navitoclax can be skipped (line 462-468), as this is not relevant for most pediatric hemato-oncologists in clinical practice.

The last three sub-sections on BTK inhibitors, proteasome inhibitors and Temsirolimus are concise and summarize the current literature data on the use of these compounds for NHL in children and adults.

I suggest that the authors should include a section on ALK inhibitors such as crizotinib or ceretinib, which are relevant for refractory/relapsed pediatric anaplastic large cell lymphoma (ALCL).

Author Response

Response to Reviewer 1 Comments:

Dear Sir or Madam, thank you very much for the review our manuscript entitled: The New Treatment Methods for Non-Hodgkin Lymphoma in Pediatric Patients.

In response to your comment, we would like to thank you for appreciating our manuscript.

Comment 1

The review written by Derebas and colleagues tries to gives an overview on contemporary treatment strategies for children and adolescents with non-Hodgkin lymphoma (NHL) with a special focus on novel treatment concepts and targeted therapies based on immunologically and genetically defined subtypes. The manuscript covers relevant aspects of contemporary treatment for pediatric NHL; however, it has major limitations: For a better reading, I first suggest to shorten and re-work the text substantially.

The introduction section contains detailed information on NHL in general, which most readers with a pediatric hemato-oncological background are usually familiar with. I suggest to shorten this part considerably. In line 40 they write “Boys develop HNL three times less often than girls, but in adolescents …, the disease of girls prevails (Ref 6)”. I cannot check reference 6, as this is an e-book in polish language, but I am sure that this statement is not true. In general, a male preponderance is reported in the literature for pediatric NHL (Burkhart, Brit J Haematol 2005 and Leukemia 2011; Reedijk, Eur J Cancer 2022).

Revision and my comment

The style of manuscript has been changed.

General, the introduction of the manuscript has been organized and condensed.

References were added (number 2 and 5).

Comment 2

In the second section (NHL treatment), they give an overview on the current treatment strategies for NHL and discuss well-known disadvantages of chemotherapy, radiotherapy and stem cell transplantation. Next, the rationale for alternative drugs or targeted therapies according to the different NHL subgroups is explained. As overview, Table 2 is provided. Table 2 contains a few misspellings such as “rituksymab” and “ibrutynib”– this needs to be corrected accordingly.

Revision and my comment

In the Table 2 the words were corrected.

Comment 3

In the upcoming section, different therapies are addressed. The sub-sections for monoclonal antibodies and antibody-drug conjugates are well written and straight to the point. The abbreviation Bv in line 143 needs to be defined (Brentuximab vedotin).

Revision and my comment

The abbreviation Bv in line 143 was defined (Brentuximab vedotin). Line 442, page 7.

Comment 4

The next sub-section on CAR-T cell therapy (starting at line 212) is much too long. I think, that explaining detailed signalling mechanisms and the evolution of different CAR-T generations is beyond the scope of this review. Instead, I suggest focusing on CAR-T cell targets relevant for pediatric NHL subtypes and to discuss experimental data and results available from clinical trials.

Revision and my comment

The section on CAR-T was shortened. The information was added – line 698-707, page 11.  

Comment 5

Furthermore, the sub-sections on EZH2 and IDH inhibitors both should be shortened substantially.

Revision and my comment

The sub-sections on EZH2 and IDH inhibitors were shortened.

Comment 6

In the next sub-section, starting at line 428, the authors explain the role of BCL-2 associated proteins in apoptotic signalling very much in detail, which is not necessary and again beyond the scope of this review. The authors should try to summarize, what information on BCL-2 is essential with respect to the treatment of NHL subtypes and which inhibitors are currently available or tested in clinical trials. In addition, they must discuss their limitations and provide clinical data, as done nicely for venetoclax. However, the evolution of these compounds from ABT-373 to navitoclax can be skipped (line 462-468), as this is not relevant for most pediatric hemato-oncologists in clinical practice.

Revision and my comment

The next section was shortened according to the review suggestion and additional information was added. Line 1343-1348, page 17; line 1361-1403, pages 17-18. 

Comment 7

The last three sub-sections on BTK inhibitors, proteasome inhibitors and Temsirolimus are concise and summarize the current literature data on the use of these compounds for NHL in children and adults.

Revision and my comment

Thank you very much for your comment.

Comment 8

I suggest that the authors should include a section on ALK inhibitors such as crizotinib or ceretinib, which are relevant for refractory/relapsed pediatric anaplastic large cell lymphoma (ALCL).

Revision and my comment

The information was added and placed in new paragraph line 1404-1427 page 18

The manuscript was also checked for English language by MDPI Editing Service. The certificate has been attached below.

We do honestly hope that it will satisfy you and improve the quality of our work. Once again, we are very grateful for your review and remain open if you have any other remarks or suggestions that will make our work merit publication in „Cancer”.

Reviewer 2 Report

There are a number of inaccurate statements in the article. While the review is well referenced, cited references often do not match the statements made. The article needs to focus more on its target which is non-Hodgkin’s lymphoma in the pediatric patients. The manuscript needs to be reorganized, re-written in  better English and statements and references carefully checked to assure accuracy.

Author Response

Response to Reviewer 1 Comments:

Dear Sir or Madam, thank you very much for the review our manuscript entitled: The New Treatment Methods for Non-Hodgkin Lymphoma in Pediatric Patients.

In response to your comment, we would like to thank you for appreciating our manuscript.

Comment 1

There are a number of inaccurate statements in the article. While the review is well referenced, cited references often do not match the statements made. The article needs to focus more on its target which is non-Hodgkin’s lymphoma in the pediatric patients. The manuscript needs to be reorganized, re-written in better English and statements and references carefully checked to assure accuracy.

Revision and my comment

The style of manuscript has been changed.

The references were checked and corrected. 

The manuscript has been organized and condensed.

The manuscript was also checked for English language by MDPI Editing Service. The certificate has been attached below.

We do honestly hope that it will satisfy you and improve the quality of our work. Once again, we are very grateful for your review and remain open if you have any other remarks or suggestions that will make our work merit publication in „Cancer”.

Reviewer 3 Report

In the present work, the authors provide the comprehensive information on various strategies for treatment of non-Hodkin lymphoma (NHL). The review will be of interest to clinical hematologists/oncologists, researchers, and students in the field of medicine and biomedicine, pharmacology and drug development. However, there are some issues to consider to improve the format of the presentation.

  1. What, if any, is the fundamental difference in lymphomagenesis, genetic/types profile, clinics and outcome in pediatric and adult patients? Are there more treatment-resistant cases in children? And, therefore, what is the difference in therapeutic strategies? What is the specifics of strategic approaches to treatment of NHL in pediatric patients?
  2. The text of the review is quite long and monotonous. The introduction of more illustrations and diagrams would greatly improved comprehension. As an example, the figures illustrating the mechanism of action of discussed drugs. Or, as another example, the figure comparing the mechanisms of naked monoclonal, Bispecific and drug-conjugated antibodies, with indication of the approved Abs in each group, and Abs in preclinical/ clinical research.

Author Response

Response to Reviewer 1 Comments:

Dear Sir or Madam, thank you very much for the review our manuscript entitled: The New Treatment Methods for Non-Hodgkin Lymphoma in Pediatric Patients.

In response to your comment, we would like to thank you for appreciating our manuscript.

Comment 1

In the present work, the authors provide the comprehensive information on various strategies for treatment of non-Hodkin lymphoma (NHL). The review will be of interest to clinical hematologists/oncologists, researchers, and students in the field of medicine and biomedicine, pharmacology and drug development. However, there are some issues to consider to improve the format of the presentation.

What, if any, is the fundamental difference in lymphomagenesis, genetic/types profile, clinics and outcome in pediatric and adult patients? Are there more treatment-resistant cases in children? And, therefore, what is the difference in therapeutic strategies? What are the specifics of strategic approaches to treatment of NHL in pediatric patients?

Revision and my comment

The information was included in the introduction section and NHL treatment in pediatric patient’s section. Line 122-123, page 3; line 135-143, page 3; line 148-158, page 3; line 293-294, page 4.

Comment 2

The text of the review is quite long and monotonous. The introduction of more illustrations and diagrams would greatly improved comprehension. As an example, the figures illustrating the mechanism of action of discussed drugs. Or, as another example, the figure comparing the mechanisms of naked monoclonal, Bispecific and drug-conjugated antibodies, with indication of the approved Abs in each group, and Abs in preclinical/ clinical research.

Revision and my comment

We added 5 additional figures.

The manuscript was also checked for English language by MDPI Editing Service. The certificate has been attached below.

We do honestly hope that it will satisfy you and improve the quality of our work. Once again, we are very grateful for your review and remain open if you have any other remarks or suggestions that will make our work merit publication in „Cancer”.

Round 2

Reviewer 1 Report

The introduction was revised by the authors and now comes with a different structure. However, I still consider this section too long as the reader gets lost in a rather unstructured mixture of epidemiologic, biologic and genetic information on NHL in general. I suggest to focus on pediatric NHL and what is known in that field so far.

Most sub-sections were successfully reworked as suggested by the reviewers.

However, the sub-section on BCL-2 inhibitors is still very long. I recommend to focus primarily on data obtained from clinical trials in the field of pediatric NHL, which is the primary aim of this manuscript.

Author Response

Response to Reviewer 1 Comments:

Dear Sir or Madam, thank you very much for the review our manuscript entitled: The New Treatment Methods for Non-Hodgkin Lymphoma in Pediatric Patients.

In response to your comment, we would like to thank you for appreciating our manuscript.

The manuscript by Justyna Derebas et al is designed to review the latest treatment options for treatment of pediatric patients with non-Hodgkin Lymphoma. While the article is well referenced, it is written in very poor English, contains a significant number of inaccuracies, a vast amount of unrelated subjects and adult-related non-pediatric data. The manuscript is poorly written and contains numerous inaccurate statements. The contents of the article are inappropriately organized, and the subjects discussed often do not flow well and are not written in a desirable and correlating fashion. The descriptions given often move from one subject to an unrelated topic.

- The authors’ statements frequently are either incorrect or do not correlate with or are not supported by the cited reference. A few examples for these criticisms are as follows:

Comment 1

In the “Simple Summary” section, the authors state that “Non-Hodgkin lymphoma is the most occurring hematologic disease in the world”. This statement is incorrect.

Revision and my comment

Our oversight has been corrected to “Non-Hodgkin lymphoma is one of the most occurring hematologic diseases in the world”

Comment 2

On line 97, the authors state that “High-dose chemotherapy with autologous stem cell transplantation (auto-SCT) was the leading therapy for younger patients with NHL, yet it resulted in high rate of relapse [20]” The statement is incorrect and unrelated to the reference cited which is regarding retrospective comparison of the outcomes of patients with relapsed/refractory non-Hodgkin lymphoma (NHL) who have undergone stem cell transplantation (SCT) with stable disease following a combination of lenalidomide and rituximab (LR) treatment and a group of patients who had not undergo SCT in a phase I/II clinical trial. [Cai, Q., Chen, Y., Zou, D., Zhang, L., Badillo, M., Zhou, S., ... & Wang, M. (2014). Clinical outcomes of a novel combination of lenalidomide and rituximab followed by stem cell transplantation for relapsed/refractory aggressive B-cell non-Hodgkin lymphoma. Oncotarget, 5(17), 7368.]

Revision and my comment

This sentence has been removed from the text.

Comment 3

The authors state that “NHL is most likely to evolve from mature B lymphocytes (B-NHL) (86%) but it can also develop from T lymphocytes T-NHL) (12%) or natural killer (NK) cells (2%)” [Reference 5, Cai, W., Zeng, Q., Zhang, X., &Ruan, W. (2021). Trends analysis of non-Hodgkin lymphoma at the national, regional, and global level, 1990–2019: results from the Global Burden of Disease Study 2019. Frontiers in medicine, 1609]. Of course, the reference does not support or correspond to the authors’ statement.

Revision and my comment

This sentence has been removed from the text.

Comment 4

Table 1 needs expansion and reorganization.

Revision and my comment

Two columns have been added to the table. They contain information about frequency of occurrence of each NHL type in pediatric NHLs and genetic rearrangements present in each type.

Comment 5

The manner in which the references are written is highly unconventional.

Revision and my comment

The manner in which the references were written has been corrected. All references are now stated individually after the sentence or at the top of the table they were used.

Comment 6

While the title of the article suggests that it is intended to discuss new treatment methods for non-Hodgkin Lymphoma in pediatric patients, much of the manuscript discusses the use of experimental therapy in other cancers and in adults.

Revision and my comment

We removed confusing sentences regarding adult treatment and other illnesses. In some sentences we briefly mention therapeutic agents for adult treatment, which are going to be tested for pediatric patients in the near future.

The manuscript was also checked for English language by MDPI Editing Service. The certificate has been attached below.

We do honestly hope that it will satisfy you and improve the quality of our work. Once again, we are very grateful for your review and remain open if you have any other remarks or suggestions that will make our work merit publication in„Cancer”.

Reviewer 2 Report

  • The manuscript by Justyna Derebas et al. is designed to comprehensively review of the novel therapies for non-Hodgkin lymphoma in the pediatric age group. The article has been extensively revised and several references have been added to the prior version. While the new revision has significantly improved the quality of the manuscript, several of the criticisms made for the last review still remains applicable.
  • The title of the article indicates that the review is regarding treatment of non-Hodgkin lymphoma in the pediatric age group. However, many parts of the review repeatedly admix the treatment of adults with the pediatric age group. This can be confusing to the reader. The manuscript also reviews the effects of several therapeutic agents in solid tumors, such as myeloma (example, line 1259), Hodgkin’s lymphoma (example, line 1393), advanced renal cell carcinoma (example, line 1414) and others, which are not related to the subject of the article.
  • While the article is an extensive review of the subject, it does not follow a proper pattern and sequence in each section, moving from one subject to another. An example is the discussion of sign and symptoms under the title of “NHL treatment in pediatric patients” (example, page 5, line 243). The mixture of the old and new version of the manuscript has exacerbated this matter.
  • Many sentences are non-specific and difficult to comprehend (example, lines 888 and 1448-1451)
  • While the article now is written in a better English language, it still requires an extensive revision (example, lines 384-385, 416-417, 498,645, 779-781, 798-800, 864-865, 888 and many others). Some sentences are non-specific (example, line 1448).
  • In the new version, the references are much more accurately cited, however, in some areas the subject of reference is not clear (example line 719, reference 66, the article was written in the People Republic of China, but the country where drug was approval is not stated).
  • Abbreviations are not always given prior to their use. A table for all abbreviations used can be helpful.

Author Response

Response to Reviewer 2 Comments:

Dear Sir or Madam, thank you very much for the review our manuscript entitled: The New Treatment Methods for Non-Hodgkin Lymphoma in Pediatric Patients.

In response to your comment, we would like to thank you for appreciating our manuscript.

The manuscript needs to be vastly altered, re-written in better English, statements and references carefully checked to assure accuracy, in order to be accepted for publication in the Journal of Cancers.The manuscript by Justyna Derebas et al. is designed to comprehensively review of the novel therapies for non-Hodgkin lymphoma in the pediatric age group. The article has been extensively revised and several references have been added to the prior version. While the new revision has significantly improved the quality of the manuscript, several of the criticisms made for the last review still remains applicable.

Comment 1

The title of the article indicates that the review is regarding treatment of non-Hodgkin lymphoma in the pediatric age group. However, many parts of the review repeatedly admix the treatment of adults with the pediatric age group. This can be confusing to the reader. The manuscript also reviews the effects of several therapeutic agents in solid tumors, such as myeloma (example, line 1259), Hodgkin’s lymphoma (example, line 1393), advanced renal cell carcinoma (example, line 1414) and others, which are not related to the subject of the article.

Revision and my comment

We removed confusing sentences regarding adult treatment and other illnesses. In some sentences we briefly mention therapeutic agents for adult treatment, which are going to be tested for pediatric patients in the near future.

Comment2

While the article is an extensive review of the subject, it does not follow a proper pattern and sequence in each section, moving from one subject to another. An example is the discussion of sign and symptoms under the title of “NHL treatment in pediatric patients” (example, page 5, line 243). The mixture of the old and new version of the manuscript has exacerbated this matter.

Revision and my comment

We reorganized our manuscript. The sign and symptoms were removed from mentioned part and now are available and properly described in Table 1.

Comment 3

Many sentences are non-specific and difficult to comprehend (example, lines 888 and 1448-1451)

Revision and my comment

Manuscript has undergone English language editing by MDPI, hence all the difficulties and mistakes were corrected.

Comment 4

While the article now is written in a better English language, it still requires an extensive revision (example, lines 384-385, 416-417, 498,645, 779-781, 798-800, 864-865, 888 and many others). Some sentences are non-specific (example, line 1448).

Revision and my comment

Our work was thoroughly revised and improved in the matter of transparency and grammatical accuracy.

Comment 5

In the new version, the references are much more accurately cited, however, in some areas the subject of reference is not clear (example line 719, reference 66, the article was written in the People Republic of China, but the country where drug was approval is not stated).

Revision and my comment

Mentioned section and citation was removed from our manuscript. We checked all of citations and we believe that currently all necessary data is given in the text.

Comment 6

Abbreviations are not always given prior to their use. A table for all abbreviations used can be helpful.

Revision and my comment

We added the table for all abbreviations and ensured that all used abbreviations in text and tables are followed by their definition.

The manuscript was also checked for English language by MDPI Editing Service. The certificate has been attached below.

We do honestly hope that it will satisfy you and improve the quality of our work. Once again, we are very grateful for your review and remain open if you have any other remarks or suggestions that will make our work merit publication in„Cancer”.

Reviewer 3 Report

The presentation of the data has improved significantly, so now I recommend MS for publication.

Author Response

Thank you for your comments very much.

Best regards

Monika Lejman